# Solving the Gross-Pitaevskii Equation with Quantic Tensor Trains: Ground States and Nonlinear Dynamics

Qian-Can Chen[1], I-Kang Liu[2], Jheng-Wei Li[3], and Chia-Min Chung[1,4,5*],

**1** Department of Physics, National Sun Yat-sen University, Kaohsiung 80424, Taiwan
**2** School of Mathematics, Statistics and Physics, Newcastle University, Newcastle upon Tyne, NE1 7RU, United Kingdom
**3** Université Grenoble Alpes, CEA, Grenoble INP, IRIG, Pheliqs, F-38000 Grenoble, France
**4** Physics Division, National Center for Theoretical Sciences, Taipei 10617, Taiwan
**5** Center for Theoretical and Computational Physics, National Sun Yat-Sen University, Kaohsiung 80424, Taiwan

⋆ chiaminchung@gmail.com

## Abstract

We develop a tensor network framework based on the quantic tensor train (QTT) format to efficiently solve the Gross-Pitaevskii equation (GPE), which governs Bose-Einstein condensates under mean-field theory. By adapting time-dependent variational principle (TDVP) and gradient descent methods, we accurately handle the GPE's nonlinearities within the QTT structure. Our approach enables high-resolution simulations with drastically reduced computational cost. We benchmark ground states and dynamics of BECs–including vortex lattice formation and breathing modes–demonstrating superior performance over conventional grid-based methods and stable long-time evolution due to saturating bond dimensions. This establishes QTT as a powerful tool for nonlinear quantum simulations.

# 1 Introduction

Efficient numerical representations of high-dimensional data are essential in both quantum many-body physics and applied mathematics [1, 2]. Tensor network methods provide a systematic approach to compressing such data while preserving essential correlations. Matrix product states (MPS), in particular, have been widely used to approximate low-entanglement quantum states, enabling accurate simulations of low-dimensional quantum systems [3, 4]. In applied mathematics, an analogous approach is the tensor train (TT) decomposition, which shares the same mathematical structure as MPS and is widely used for efficiently representing high-dimensional data [5–7].

A key extension of tensor train decomposition is the quantic tensor train (QTT), a quantum-inspired numerical method that efficiently discretizes continuous functions on an exponential grid. A QTT maps high-dimensional tensors into a logarithmic-size tensor train, which significantly reduces storage requirements and computational complexity from exponential to polynomial [8,9]. Unlike conventional grid-based discretization techniques, QTT exploits a hierarchical structure similar to quantum wavefunction representations, where different "qubits" correspond to different length scales. Consequently, the tensor ranks for smooth functions are typically small. The ability to encode large-scale problems makes QTT powerful for various applications, including Fourier transforms [10,11], quantum field theory [12–14], partial differential equations [15, 16], quantum chemistry [17], and data compression [18,19].

Since QTT can efficiently represent continuous-space functions and operators, it is particularly useful for solving quantum systems involving single particles or systems under the mean-field approximations [20, 21]. Given that MPS and QTT essentially share the same tensor network structure but differ in their applications, numerical algorithms developed for many-body wavefunctions, such as the density matrix renormalization group (DMRG) [3,4], can be applied naturally to QTT wavefunctions. Similarly, time evolutions of QTT wavefunctions can be effectively performed using the time-dependent variational principle (TDVP) [22,23].

In this work, we are particularly interested in the applications of solving the Gross-Pitaevskii equation (GPE) [24], a non-linear Schrödinger equation describing behaviors of Bose-Einstein condensates (BECs), where a large fraction of particles condense onto a single quantum state [25], under the mean-field approximation. BECs have been experimentally realised in the highly tunable ultra-cold dilute gas systems in a variety of settings with the state-of-the-art experimental technologies, e.g. [26–37]. This allows researchers

to study the dynamics of macroscopic wavefunction, superfluidity, collective excitations, topological excitations, superfluid turbulences and etc. [28], which have been successfully captured by the GPE. BEC systems are closely related superfluidity, and the emergence of quantum vortices is one of the hallmarks of that. Vortices are the angular momentum carriers in superfluid systems and can be generated by, for example, rotation [29], external stirring [32] or atom-light coupling [35]. For large-scale systems accommodating multiple vortices, accurate numerical simulations require sufficiently fine spatial discretisation, presenting significant computational challenges. Similar difficulties arise in large-scale simulations of nonlinear Schrödinger equations in cosmological [38–41] and astrophysical systems [42–44]. The QTT framework effectively addresses these issues by enabling an exponential increase in discretization points.

Due to the intrinsic nonlinearity of the GPE, standard MPS methods, such as DMRG and TDVP, cannot be applied directly without modifications. We extend existing MPS algorithms to incorporate the nonlinearities of the GPE, allowing us to efficiently compute its ground states and dynamical evolution. To find the ground states, we consider two numerical approaches: imaginary-time evolution (ITE), also called as the imaginary-time propagation method [45], and variational methods. We benchmark our methods by considering a cylindrically harmonically trapped BEC in two dimensions with and without rotation for ground-state solutions and the dynamics of its breathing-mode (mono-pole) excitation. Rotating a BEC results in the formation of quantum vortices, which organise themselves into a triangular lattice format, similar to the Abrikosov lattice in superconductors [46]. On the other hand, the breathing-mode excitation is a BEC width oscillation with a trapping-frequency-dependent frequency for a sufficiently large BEC [47–49], and we generate it by a sudden quench in the interaction strength.

Our results show that QTT methods significantly speed up calculations compared to conventional methods with grid-based discretization, especially for large discretizations. We show that, between the two ground-state methods, the variational approach is more efficient than ITE. We demonstrate that, in contrast to MPS time evolution for many-body quantum states—where the bond dimension typically grows exponentially with time—the bond dimension in the QTT representation saturates at long times. This property allows for efficient and stable simulations of long time scales.

We note that similar ideas have been explored independently in Ref. [50–52], which appeared concurrently with our work.

The structure of the paper is as follows. In Sec. 2, we define the physical systems. In Sec. 3, we explain the numerical techniques used to find ground states. Benchmark results are presented in Sec. 4. Finally, we summarize our conclusions in Sec. 5. In the Supplementary Material, we review the basic concepts of QTT and explicitly show the tensors used in the Hamiltonian.

## 2   Gross-Pitaevskii equation

GPE is a nonlinear Schrödinger equation describing the weakly interacting BEC under mean-field approximation. In this work, we consider BEC in a rotating harmonic trap [53]. The Hamiltonian is given by

$$\hat{H} = \hat{H}^1 + \hat{H}_{\text{int}}, \tag{1}$$

$$\hat{H}^1 = -\frac{\hbar^2}{2m}\nabla^2 + \frac{1}{2}m\omega^2 r^2 - \Omega\hat{L}_z, \tag{2}$$

$$\hat{H}_{\text{int}} = g|\psi(\mathbf{r}, t)|^2 \tag{3}$$

where $\hat{H}^1$ is the one-body Hamiltonian including the kinetic energy, the harmonic trapping potential $\frac{1}{2}m\omega^2 r^2$, and the rotational term $\Omega\hat{L}_z$. $\hat{H}_{\text{int}}$ is the interaction with strength $g = 4\pi\hbar^2 a_s N/m$, where $a_s$ is the $s$-wave scattering length, $N$ is the total particle number and $m$ is the particle mass. Throughout this work, we consider the dimensionless Hamiltonian by adopting the characteristic scales of quantum harmonic oscillator units, $\sqrt{\hbar/m\omega}$, $1/\omega$ and $\hbar\omega$ for the length, time and energy respectively (This is equivalent to set $\hbar = 1$, $m = 1$, and $\omega = 1$). The condensate wavefunction is normalized to one. The evolution of the wavefunction is governed by the time-dependent GPE

$$i\hbar\frac{\partial\psi(\mathbf{r}, t)}{\partial t} = \hat{H}\psi(\mathbf{r}, t) \tag{4}$$

For stationary states, the wavefunction takes the form $\psi(\mathbf{r}, t) = \psi(\mathbf{r})e^{-i\mu t/\hbar}$, where $\mu$ is the chemical potential. This yields the time-independent GPE

$$\mu\psi(\mathbf{r}) = \hat{H}\psi(\mathbf{r}). \tag{5}$$

The ground state is defined by minimizing the energy per particle,

$$E[\psi] = E^1[\psi] + E_{\text{int}}[\psi] \quad \text{with} \tag{6}$$

$$E^1[\psi] = \int d^3\mathbf{r}\,\psi^*\hat{H}^1\psi(\mathbf{r}) \; ; \tag{7}$$

$$E_{\text{int}}[\psi] = \frac{g}{2}N\int d^3\mathbf{r}|\psi(\mathbf{r})|^4. \tag{8}$$

The interaction energy is quartic in the wavefunction due to the term $|\psi(\mathbf{r})|^4$, which introduces nonlinearity. The factor $\frac{1}{2}$ in Eq. (8) avoids double counting the interactions.

# 3 Methods

In this section, we introduce two methods for computing the ground state of the GPE, including the ITE and a gradient descent method. For real- or imaginary-time evolution, TDVP is a powerful method with the MPS/QTT framework, and GD is effective for variationally optimizing the ground state. Here, we explain the necessary modifications to both methods to handle the nonlinear interaction term in the Hamiltonian.

## 3.1 Imaginary-time evolution: TDVP

ITE is a common strategy for computing the ground state of the GPE [45] and can be applied to the QTT wavefunction with minor modifications. Given the wavefunction (Hamiltonian) as a QTT (QTTO), the time evolution of a small time step $\delta\tau$ can be performed by using TDVP algorithm [22, 23]. Although the Hamiltonian in QTTO representation is typically "long-range" in the sense that it couples qubits over large distances, TDVP operates directly on the QTTO Hamiltonian, which makes it particularly well suited for

QTT time evolutions. The Hamiltonian of the GPE contains the one-body term $\hat{H}^1$ and the interaction term $\hat{H}_{\text{int}}$. The interaction term $gN|\psi(\mathbf{r})|^2$ depends on the wavefunction $\psi(\mathbf{r})$, and therefore needs to be updated at each step using the evolved wavefunction from the last step. The time step $\delta\tau$ is set to be $0.5\delta x^2$ to satisfy the Courant-Friedrichs-Lewy condition and ensure the stability of the convergence.

An essential step in the TDVP algorithm is to compute the effective Hamiltonian, where the required tensor contractions are shown in Fig. 1(a). The computational cost is proportional to $dD^3D_W + d^2D^2D_W^2$, where $d = 2$ is the dimension for each qubit, and $D$ and $D_W$ are the bond dimensions of the wavefunction and the Hamiltonian. We keep $\hat{H}^1$ and $\hat{H}_{\text{int}}$ as two QTTOs and compute their effective Hamiltonians separately. For $\hat{H}^1$, the bond dimension is fixed and small, and the corresponding computational cost is relatively low. However, the QTTO for the interaction has a bond dimension $D^2$ for an exact calculation, where $D$ is the bond dimension of the wavefunction. Since $D$ controls the accuracy of the wavefunction and can be large in a challenging system, we need to compress $\psi^2(\mathbf{r})$ from a bond dimension $D^2$ to a smaller bond dimension $\chi$ to achieve efficient computations (see Fig. 1(b)). The computational cost then depends on the efficiency of this compression. In this work, we perform the compression using the variational method by maximizing the overlap $\langle\psi^2|\psi_\chi^2\rangle$ [2], where $\psi_\chi^2(\mathbf{r})$ is the compressed $\psi^2(\mathbf{r})$ with the bond dimension $\chi$. In practice, $\psi^2(\mathbf{r})$ typically has a structure similar to that of $\psi(\mathbf{r})$, and therefore $\chi$ is typically on the same order as $D$. This yields an overall computational complexity of roughly $O(D^4)$.

## 3.2 Variational method: gradient descent

Another powerful approach to the ground state is to use the variational method to minimize the energy $E[\psi]$. Although DMRG is not applicable due to the nonlinearity of the Hamiltonian, gradient descent remains a viable optimization method. Gradient descent has been used to optimize an MPS when the tensors are isometry tensors [54–57]. In this work, we employ gradient descent within the MPS/QTT framework. The algorithm proceeds as follows. The QTT wavefunction is represented in canonical form, with one tensor at the orthogonality center–initially at the first site–and all tensors to the left (right) being left (right) orthogonal. The tensor at the orthogonality center is updated based on its energy gradient until it converges to a minimum. Once optimized, the orthogonality center is moved to the next site, and the process is repeated. This sweep continues back and forth until the global energy minimum is reached.

An essential step in the gradient descent method is to compute the energy gradient with respect to a tensor in the QTT wavefunction. The computational complexity is proportional to $dD^3D_W + d^2D^2D_W^2$. Again, we compute the gradients of the one-body energy $E^1[\psi]$ and the interaction energy $E_{\text{int}}[\psi]$ separately. Similarly to the ITE, the gradient of $E^1[\psi]$ can be computed straightforwardly with relatively low cost, but the exact computation for the gradient of $E_{\text{int}}[\psi]$ has complexity proportional to $D^5$ and is very slow (see Fig. 1(c)). To achieve efficient calculations, we compress $\psi^2(\mathbf{r})$ to a smaller bond dimension $\chi$. The computation of the gradient $\partial_A E_{\text{int}}[\psi]$ is then reduced to a cost proportional to $D^4$, which is the same as in the ITE.

There is one thing we need to pay additional attention to in this approach. Since $\psi_\chi^2(\mathbf{r})$ is a single QTT that represents the product of wavefunctions $\psi^2(\mathbf{r})$, changing one tensor in $\psi(\mathbf{r})$ in principle changes *all* the tensors in $\psi_\chi^2(\mathbf{r})$. This means that *all* the tensors in $\psi_\chi^2(\mathbf{r})$ needed to be recomputed for *each* update of a local tensor in $\psi(\mathbf{r})$, even in the internal loops of gradient descent. This significantly slows down the computations. Fortunately, in practice, we found that updating a few tensors around the orthogonality site in $\psi_\chi^2(\mathbf{r})$ is sufficient to achieve high accuracies for the gradient $\partial_A E_{\text{int}}[\psi]$ and for the

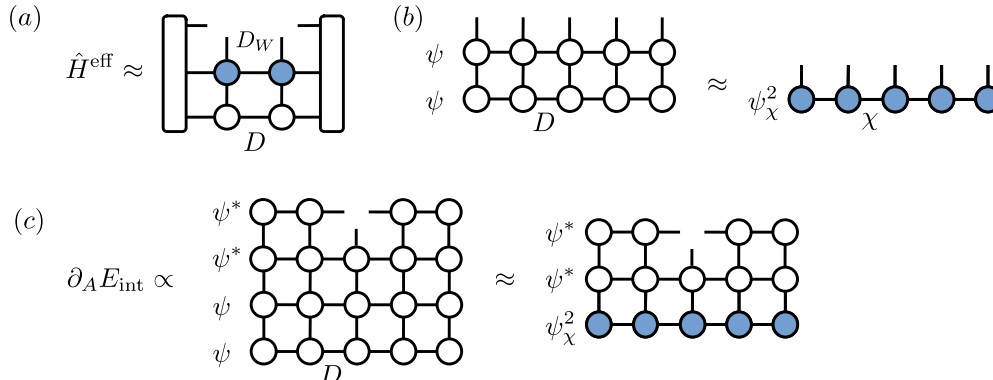

Figure 1: Schematic diagrams of tensor contractions and QTT structures. (a) The tensor contractions for computing the effective Hamiltonian. (b) QTT wavefunctions for $\psi^2$ and its compressed QTT form $\psi_\chi^2$. (c) Tensor contractions involved in computing the gradient of the interaction energy, shown for both the exact calculation and the approximation using $\psi_\chi^2$.

slope along the descent direction. This is reasonable because in each update the tensor is changed only slightly, and the "correlation length" (regarding the qubits rather than the real space) is restricted by the bond dimension. Therefore, when the bond dimensions are small, changing a tensor in $\psi(\mathbf{r})$ corresponds to a "short-range" update for $\psi^2(\mathbf{r})$. As a demonstration, Fig. 2 shows the errors of the gradient $\partial_A E_{\text{int}}[\psi]$ and the slope along the descent direction, plotted as functions of the number of updating tensors in $\psi_\chi^2(\mathbf{r})$. The system is a BEC in a harmonic trap. It can be seen that the errors drop below $10^{-9}$ ($10^{-14}$) for the gradient (slope) when only three tensors are updated. In this work, we choose to update three tensors in $\psi_\chi^2(\mathbf{r})$ when a tensor in $\psi(\mathbf{r})$ is updated.

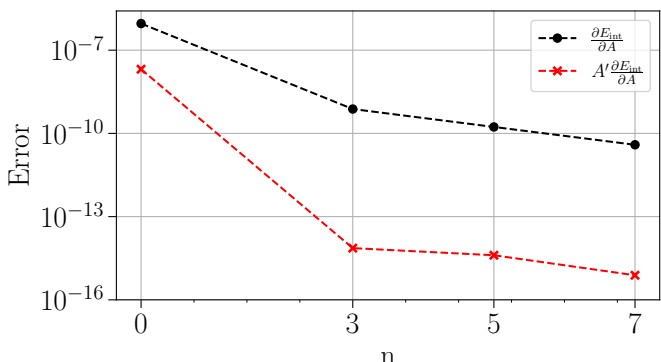

Figure 2: Errors in the gradient of the interaction energy (black curve) and in the slope along the descent direction (red curve) as functions of the number of tensors updated in $\psi_\chi^2$ after updating a tensor in $\psi$.

In this work, we employ the basic gradient descent method with line searches that satisfy the strong Wolfe conditions. While this approach is effective, more advanced optimization algorithms, such as quasi-Newton methods or the conjugate gradient descent method, can be used to further enhance convergence performance.

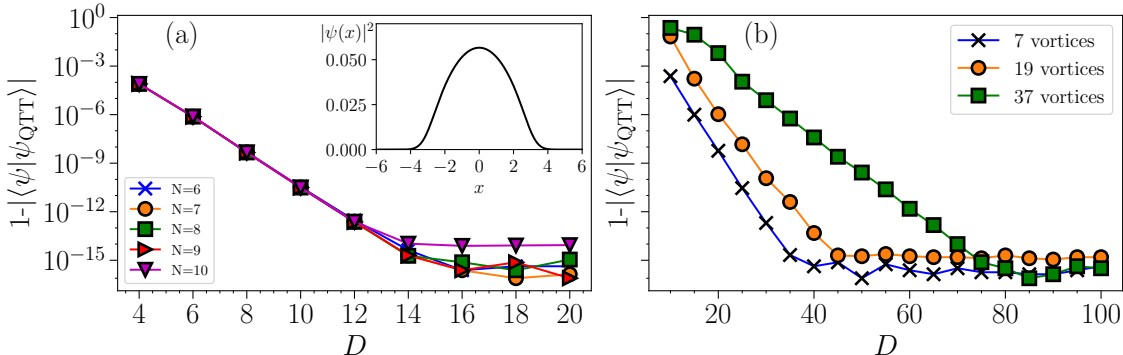

Figure 3: Errors in the QTT compression of ground-state wavefunctions for (a) a BEC in a harmonic trap and (b) a BEC with vortices, plotted as functions of bond dimension $D$. The inset in panel (a) shows the density profile of the corresponding ground-state wavefunction.

## 4   Benchmark

### 4.1   BEC in a harmonic potential

We first benchmark our method on a system of a BEC in a harmonic trap. This corresponds to $\Omega = 0$ and $g = 100$ in the Hamiltonian in Eq. 3. The particle density of the ground-state wavefunction is shown in the inset of Fig. 3(a). This function can be efficiently represented in a QTT format with small bond dimensions. In Fig. 3(a), we show the errors of the QTT approximations with different bond dimensions $D$. The error is defined by $1 - |\langle\psi|\psi_{\mathrm{QTT}}\rangle|$ where $|\psi\rangle$ is the exact ground state and $|\psi_{\mathrm{QTT}}\rangle$ is the QTT wavefunction after compression. One can see that QTT achieves machine accuracy with error $\approx 10^{-16}$ with a bond dimension $D = 20$, and the error is not sensitive to the number of qubits. We thus choose $D = 20$ in the following benchmark.

To demonstrate the efficiency of QTT in terms of the number of grid points, we compare the computational time required for a single ITE step between the QTT method and the conventional method. With QTT formats, the number of qubits increases exponentially as the number of qubits increases linearly. This is directly reflected in the computational time: When the number of grid points increases exponentially, the QTT computational time scales only linearly, while the computational time for conventional discretizations grows exponentially, as shown in Fig. 4.

We then compare the efficiencies for the two ground-state algorithms, ITE and gradient descent. In Figs. 5(a) and (b), we show the errors of the chemical potential $\mu$ as functions of the CPU time and of the optimization steps for each method. Note that the variational method minimizes the energy $E[\psi]$ rather than the chemical potential $\mu$. Therefore, $\mu$ is not variational and can be lower than the exact value during optimization. For ITE, an optimization step is defined by an evolution of a time step $\delta\tau$, and for the variational method, an optimization is defined by a complete sweep of optimizations. Both methods obtain $\mu$ with an accuracy up to $10^{-10}$ with bond dimension $D = 20$, showing excellent representations of the ground state. Although the ITE is slightly faster for an optimization step, gradient descent has a better overall efficiency because, in each iteration, it converges much faster than the ITE. For a fixed $D$, the errors saturate in a large number of optimization steps because the bond dimension $D$ limits the accuracy of $\psi(\mathbf{r})$. One can systematically improve the accuracy by increasing the bond dimension $D$.

As mentioned in Sec. 3, the computational cost significantly depends on the efficiency

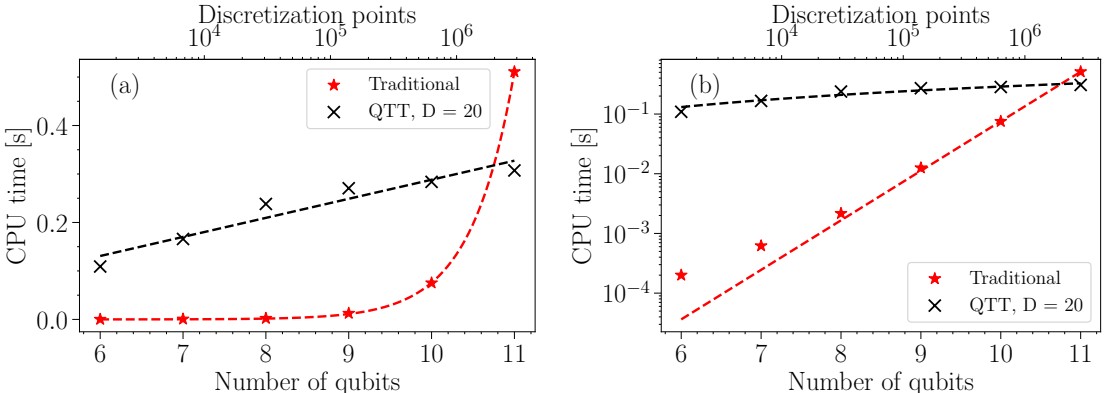

Figure 4: CPU times for a single ITE step using the QTT and conventional methods, plotted as functions of the number of qubits (equivalently, the number of grid points). Results are shown in (a) linear and (b) logarithmic scales to highlight the different scaling behaviors of the two approaches.

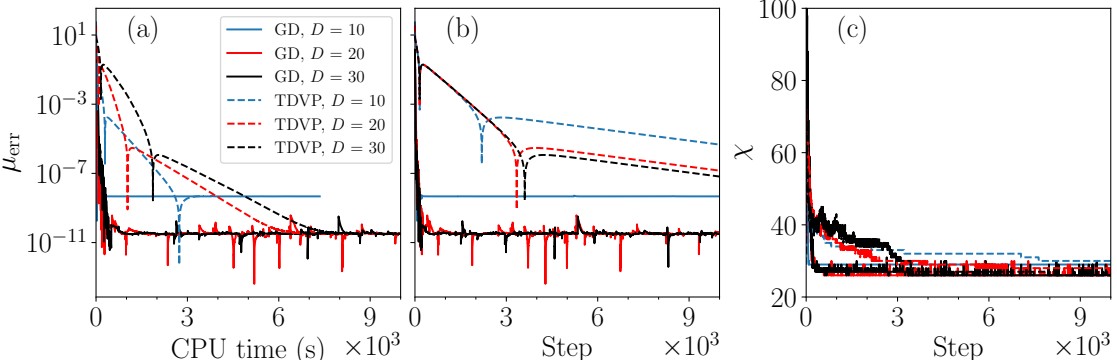

Figure 5: Convergence of the chemical potential $\mu$ computed using gradient descent (GD) and TDVP within the QTT framework, shown as functions of (a) CPU time and (b) optimization steps, for a BEC in a harmonic trap. Different colors correspond to different bond dimensions of the QTT wavefunctions. Panel (c) shows the evolution of the bond dimension of $\psi_\chi^2(\mathbf{r})$ during the optimization.

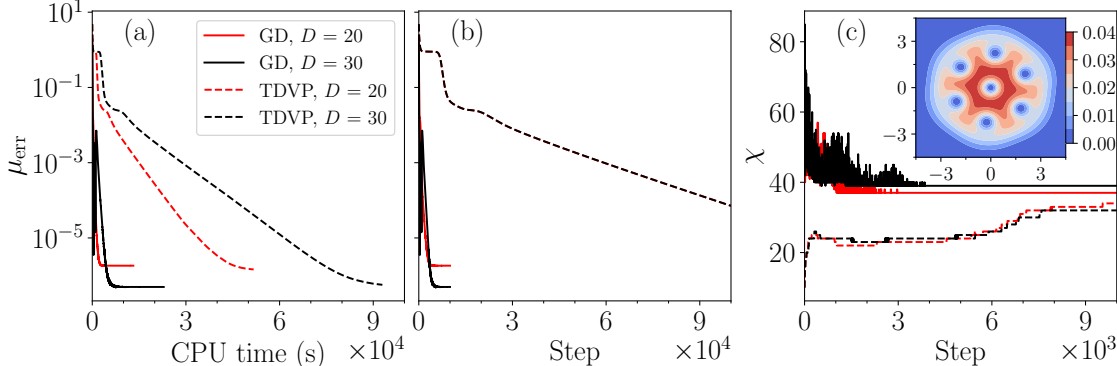

Figure 6: Convergence of the chemical potential $\mu$ computed using gradient descent (GD) and TDVP within the QTT framework, as functions of (a) CPU time and (b) optimization steps, for a BEC ground state containing 7 vortices. Different colors indicate different bond dimensions of the QTT wavefunction. Panel (c) shows the evolution of the bond dimension of $\psi_\chi^2(\mathbf{r})$ during optimization. The inset displays the corresponding density profile.

of compressing $\psi^2(\mathbf{r})$ to a smaller bond dimension $\chi$. In Fig. 5(c), we show the behaviors of $\chi$ as functions of optimization steps, given the truncation cutoff $\epsilon_{\psi^2} = 10^{-12}$. It can be seen that, for both ITE and GD, $\chi$ starts from rather large initial values (but still much smaller than $D^2$) in the early stages of the simulations, and eventually converges to values close to $D$. This can be understood because in the early stages of the simulations, the variational wavefunctions are far from the ground state and can lead to rather complicated structures in $\psi_\chi^2(\mathbf{r})$. When the wavefunction approaches the ground state, which is typically smooth, $\psi_\chi^2(\mathbf{r})$ has similar structures to $\psi(\mathbf{r})$ and the bond dimension $\chi$ saturates to a value in the same order as $D$.

## 4.2 Rotating BEC

We then consider a more challenging system, a BEC in a rotating frame with angular frequency $\Omega$. The system will induce vortices as the lowest-energy excitations, which form a triangular lattice. The number of vortices depends on the angular momentum $\Omega$ and the system size [47]. Fig. 3(b) shows the errors of QTT compressions for the ground states versus the bond dimension $D$ for different numbers of vortices and with the number of qubits $N_{\text{qub}} = 8$ (corresponding to 256 grid points) for each dimension. Compared to the system without a vortex, the wavefunctions require larger bond dimensions to achieve high accuracies, and the required bond dimensions depend on the number of vortices.

To benchmark the efficiency of the methods, we first focus on a Hamiltonian with $\Omega = 0.8$ and $g = 100$, where the ground state has 7 vortices. The vortex pattern for the ground state is shown in the inset of Fig. 6(c), showing a clear triangular lattice. We compare the computational efficiencies of the two QTT ground-state methods. Fig. 6(a)(b) shows the errors of chemical potential $\mu$ as functions of the CPU time and of the optimization steps. Similarly, the errors obtained from gradient descent decrease significantly faster than those obtained from ITE, showing the outperformance of gradient descent over ITE. The accuracies can be systematically improved by increasing the bond dimensions $D$. During gradient descent optimization, the bond dimensions $\chi$ of the compressed wavefunction square $\psi_\chi^2(\mathbf{r})$ start from relatively large values, and gradually decreases, eventually saturating to values on the same order as $D$, given a truncation cutoff of $\epsilon_{\psi^2} = 10^{-8}$, as shown in Fig. 6(c). In the ITE, in contrast, the bond dimension $\chi$ initially has lower values,

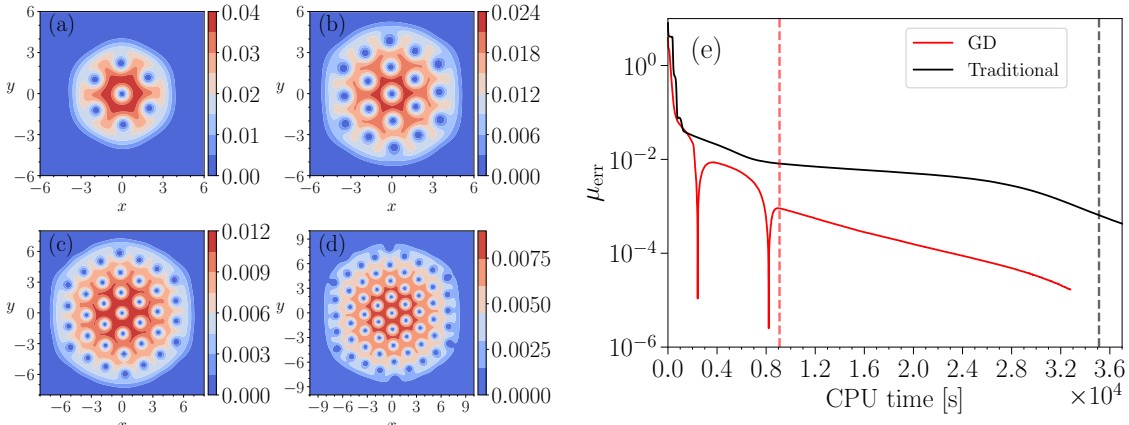

Figure 7: Density profiles of the ground states containing (a) 7, (b) 19, (c) 37, and (d) 61 vortices. Panel (e) shows the convergence of the chemical potential for the 19-vortex state, computed using the QTT-based gradient descent (GD) method and a conventional grid-based approach. Dashed lines mark the points at which well-defined triangular vortex lattices begin to emerge.

| Number of vortices | $\Omega$ | $g$ | $D$ | $N_{\text{qub}}$ |
|---|---|---|---|---|
| 7 | 0.8 | 100 | 20 | 8 |
| 19 | 0.95 | 100 | 30 | 8 |
| 37 | 0.92 | 500 | 40 | 9 |
| 61 | 0.969 | 500 | 50 | 9 |

Table 1: Hamiltonian parameters $\Omega$ and $g$, the bond dimension $D$, and the number of qubits $N_{\text{qub}}$ for each dimension for the QTT ground states containing different numbers of vortices shown in Fig. 7.

then increases, and eventually saturates to the same order as $D$ during the evolution.

We then consider several systems with larger $\Omega$, where the numbers of vortices in the ground states range from 19 to 61. We obtain well-converged wavefunctions that show clear triangular lattices in all cases. Their densities are shown in Fig. 7(a)-(d). We found that using the wavefunction with fewer vortices as the initial state significantly improves the convergence when computing ground states with a larger number of vortices. It is also helpful to start from a QTT with fewer qubits and gradually increase the number of qubits by linear interpolation. The corresponding parameters in the Hamiltonian and the bond dimensions of $\psi(\mathbf{r})$ are summarized in Tab. 1.

For the system with 19 vortices, we benchmark our gradient descent method with the ITE method with conventional discretization. We compare the errors of chemical potential $\mu$ between the two methods as functions of CPU time, shown in Fig. 7(e). It can be observed that the QTT gradient descent method converges more efficiently than the conventional method. The dashed lines indicate roughly when the well-defined triangular vortex lattices emerge in both methods. The QTT method captures the correct vortex pattern several times faster than the conventional method. We choose this system and parameter set for benchmarking in order to keep the computational time of the conventional method manageable. We expect the QTT method to outperform the conventional method even further in systems with more vortices and finer grids.

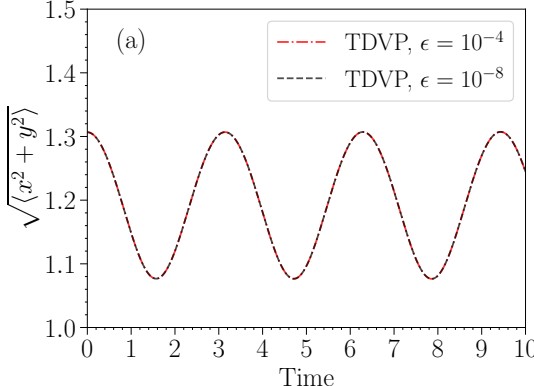 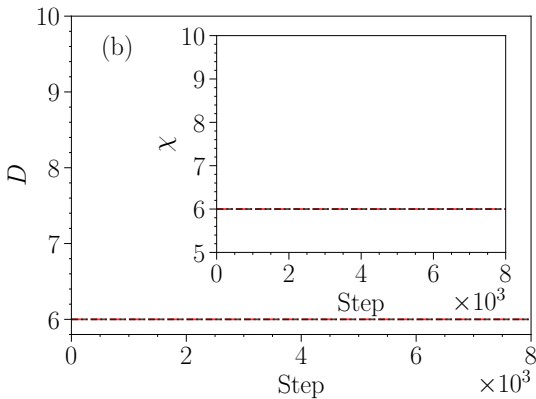

Figure 8: (a) Time evolution of the wavefunction width in the breathing mode simulated using TDVP with two different truncation cutoffs $\epsilon$. (b) Bond dimensions $D$ of $\psi(\mathbf{r})$ and $\chi$ of $\psi_\chi^2(\mathbf{r})$ as functions of optimization step for the corresponding truncation cutoffs.

## 4.3 Dynamics

The real-time evolution of a QTT wavefunction can be performed using the same TDVP algorithm employed for the ground state. Here, we consider the time evolution of a BEC in a harmonic trap ($\Omega = 0$) after a sudden quench from an initial interaction strength $g_1 \approx 12.5$ to a smaller value $g_2 = g_1/2$. After the quench, the system behaves in a so-called *breathing mode* where the width of the wavefunction oscillates in time with frequency $2\omega$ [47,58,59], where $\omega$ is the harmonic trap frequency defined in the Hamiltonian in Eq. 3. Fig. 8(a) shows the wavefunction width as a function of time for two different truncation cutoffs $\epsilon$. In both cases, the results perfectly agree with the expected oscillation frequency.

In these simulations, both the bond dimensions of $\psi(\mathbf{r})$ and $\psi^2(\mathbf{r})$ are controlled by the same cutoff $\epsilon$, and can in principle grow in time. However, there is an essential difference between the evolution of QTT wavefunctions and many-body MPS wavefunctions. For evolutions of many-body states, the MPS bond dimensions typically grow exponentially over time due to the growth of entanglement, which limits the accessible time. In contrast, the bond dimension of a QTT depends on the smoothness and the correlations between different scales of the wavefunction. If the wavefunction remains smooth during the evolution, a QTT wavefunction may have a small bond dimension for an arbitrary long time. In the systems we consider, the wavefunction remains smooth and simple throughout the entire evolution, allowing for efficient simulations over arbitrarily long times. In Fig. 8(b), we show that the bond dimensions of $\psi(\mathbf{r})$ and $\psi^2(\mathbf{r})$ as functions of time with truncation cutoff $\epsilon = 10^{-8}$. They stay with small values and do not grow over the entire simulation. These results indicate that the QTT framework is a promising approach for simulating the dynamics governed by the GPE.

## 5 Conclusion and Discussion

We have introduced an efficient and scalable computational framework based on the QTT format for solving the GPE, which models BECs within the mean-field approximation. By extending tensor network techniques, we adapted the TDVP and developed a gradient-based variational method to handle the nonlinear interaction term. A key element of our

approach is the compression of the nonlinear term into a compact QTT representation, which greatly reduces computational complexity without compromising accuracy. This allows for the simulation of large, highly resolved systems that are otherwise difficult to treat with conventional grid-based methods.

We demonstrated that QTT-based methods outperform conventional approaches in both accuracy and efficiency, particularly as the system size and resolution increase. For ground-state calculations, the gradient descent algorithm converges more rapidly and achieves better numerical performance than imaginary-time evolution. In rotating condensates, we computed vortex lattice ground states with up to several dozen vortices using modest computational resources. For real-time dynamics, we showed that the bond dimension remains stable over long simulation times, enabling efficient and accurate simulations of nonlinear quantum evolution.

Overall, our results establish the QTT framework as a powerful and versatile tool for solving nonlinear quantum equations. By combining high compression efficiency with algorithmic flexibility, it provides a path forward for large-scale simulations in mean-field quantum systems and offers new opportunities for applying tensor network methods to nonlinear and continuum problems in physics.

## Acknowledgements

We acknowledge helpful discussions with X. Waintal.

**Funding information** C.-M. C. acknowledges support by MOST (111-2112-M-110-006-MY3 and 113-2112-M-110-026) and the Yushan Young Scholar Program under the Ministry of Education (MOE) in Taiwan. I.-K. Liu is supported the Science and Technology Facilities Council (ST/W001020/1). J.-W. L. acknowledges funding support from the Plan France 2030 ANR-22-PETQ-0007 "EPIQ"

## A  Normalization Problem

The normalization consistency between continuum and discrete representations poses a fundamental challenge in quantum tensor train (QTT) simulations. Consider the 2D Gross-Pitaevskii equation (GPE) chemical potential formulation:

$$\mu_{2D} = \sum_{i,j} \psi^*(\mathbf{r}_{ij}) \left[ -\frac{1}{2}\nabla_{i,j}^2 + \frac{1}{2}\omega^2 \mathbf{r}_{ij}^2 + g|\psi(\mathbf{r}_{ij})|^2 - \Omega L_z \right] \psi(\mathbf{r}_{ij}) \Delta x \Delta y \qquad (9)$$

where $\mathbf{r}_{ij} = (x_i, y_j)$ denotes discrete grid coordinates with uniform spacing $\Delta x = \Delta y$. The normalization condition enforces:

$$\sum_{i,j} |\psi(\mathbf{r}_{ij})|^2 \Delta x \Delta y = 1 \qquad (10)$$

For the QTT canonical form, we introduce dimensionless wavefunction $\tilde{\psi}$ through the transformation:

$$\tilde{\psi}(\mathbf{r}_{ij}) = \psi(\mathbf{r}_{ij})\Delta x \qquad (11)$$

yielding modified normalization:

$$\sum_{i,j} |\tilde{\psi}(\mathbf{r}_{ij})|^2 = 1 \qquad (12)$$

The Hamiltonian undergoes corresponding rescaling:

$$\hat{H}[\tilde{\psi}] = -\frac{1}{2}\nabla^2 + \frac{1}{2}\omega^2 r^2 + \frac{g}{(\Delta x)^2}|\tilde{\psi}|^2 - \Omega L_z \tag{13}$$

The QTT chemical potential calculation preserves continuum values through careful dimensional analysis:

$$\tilde{\mu}_{2D} = \sum_{i,j} \tilde{\psi}^*(\mathbf{r}_{ij})\hat{H}[\tilde{\psi}]\tilde{\psi}(\mathbf{r}_{ij}) = \sum_{i,j} \psi^*(\mathbf{r}_{ij})\Delta x \left[\hat{H}[\psi]\right]\psi(\mathbf{r}_{ij})\Delta x = \mu_{2D}$$

The rescaled nonlinear coupling parameter exhibits critical scaling behavior:

$$g' = \frac{g}{(\Delta x)^2}. \tag{14}$$

For typical parameters $\Delta x < 10^{-8}$ and $g > 10^2$, we observe:

$$g' > 10^{18} \quad \text{(2D case)}, \quad g' > 10^{27} \quad \text{(3D case)}. \tag{15}$$

This divergence imposes severe numerical stability constraints, particularly in higher dimensions where the scaling becomes $g' \propto (\Delta x)^{-d}$ for spatial dimension $d$.

# B    Review of Quantic Tensor Train

In this section, we review the basic concepts of the quantic tensor train (QTT) used in this work. We explain how a function can be represented in the QTT format and how operators are expressed as QTT operators (QTTO).

**From function to tensor.** Since QTT is a decomposition method for multi-index tensors, it is helpful to first clarify how a function can be represented as an $N$-index tensor. For simplicity, consider a one-dimensional function $f(x)$ defined on a discretized grid with $2^N$ points $\{x_i\}$, where $i = 1, 2, \ldots, 2^N$. Each index $i$ can be represented by an $N$-bit binary number: $\{b_1 b_2 \ldots b_N\}$, with $b_j \in \{0, 1\}$. For example, $i = 1$ corresponds to $\{00\ldots00\}$, $i = 2$ to $\{10\ldots00\}$, and so forth. Each bit $b_j$ is often referred to as a *qubit*, highlighting the conceptual link to quantum computing. The discretized function values $f(x_i)$ can thus be relabeled as $f_{b_1 b_2 \ldots b_N}$, forming an $N$-index tensor, as illustrated in Fig. 9(a). This relabeling is exact and does not involve approximation; the tensor retains all $2^N$ original function values. In graphical tensor notation, this representation is illustrated in Fig. 9(b), where each open leg represents a qubit index. Once in this form, the tensor can be decomposed into the QTT format.

**TT/MPS decomposition.** The QTT decomposition represents an $N$-index tensor as a chain of $N$ smaller tensors connected through internal indices known as *bond dimensions*, as shown in Fig. 9(c). Smooth functions typically have low correlations across different scales, allowing the tensor to be compressed efficiently into a QTT form with small bond dimensions. For certain functions, exact QTT decompositions are known (see Sec. C for examples). In general, however, one must numerically determine the QTT representation using techniques such as *tensor cross interpolation* (TCI) [60–64]. TCI is a powerful algorithm that learns QTT tensors from sampled values of the target function, controlling the bond dimensions and thus the accuracy of the approximation.

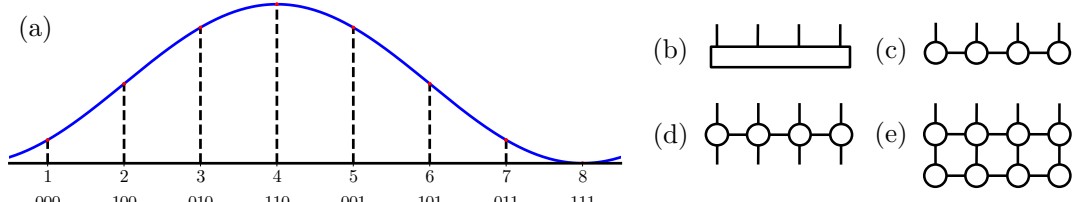

Figure 9: (a) TT/MPS representation of a 4-index tensor. The open legs represent the qubits/physical indices, and the shared legs represent the virtual indices. (b) An MPO (or tensor train operator) that represents an operator, for example $\frac{d^2}{dx^2}$. (c) MPO-MPS contraction, which computes the application of an operator to a function, for example $\frac{d^2}{dx^2}f$, or a product of two function $f \cdot g$.

For functions of two or higher dimensions, the representation can be generalized straightforwardly by assigning $N$ qubits to each spatial dimension. In this work, we specifically focus on two-dimensional wavefunctions. We use the so-called *mirror ordering*, placing qubits representing large-scale structures at the center of the QTT chain and those corresponding to small-scale details at the edges.

The computational cost grows linearly with the total number of qubits, making simulations at exponentially fine discretization straightforward. The main computational challenge typically arises from the bond dimensions, which strongly depend on the structure of the target functions and thus vary across different systems.

**Operator representation as QTTO.** Similar to the representation of functions in QTT format, physical operators can be represented in the QTT framework as matrix product operators (MPOs) by doubling the number of qubits, as illustrated in Fig. 9(d). In this work, we focus mainly on the Laplacian operator $\nabla^2$, which appears naturally in terms of kinetic energy. An exact MPO representation for this operator is given explicitly in the Supplementary Material.

**Function operations in QTT format.** Operations such as the product or addition of two functions can be performed directly in the QTT format. To calculate a product $f(x)g(x)$, it is often convenient to represent one of the functions (e.g., $f(x)$) as a QTTO. Specifically, starting from the QTT form of the function $f(x)$, each tensor's qubit index is duplicated into a diagonal form, transforming a QTT tensor $A^{b_i}_{\alpha_i \beta_i}$ into an MPO-type tensor $A^{b_i b'_i}_{\alpha_i \beta_i} = A^{b_i}_{\alpha_i \beta_i} \delta_{b_i b'_i}$, where $b_i$ denotes the qubit index and $\alpha_i, \beta_i$ represent the left and right bond indices, respectively. With $f(x)$ now in QTTO form, the product $f(x)g(x)$ can be computed efficiently through an MPO-MPS contraction, as depicted in Fig. 9(e).

Similarly, the addition of two functions or operators can be efficiently implemented using standard methods for adding MPSs or MPOs [2]. Using this approach, complex operators can be constructed directly from simpler, elementary operators. For example, the angular momentum operator $\hat{L}_z = \hat{x}\hat{p}_y - \hat{y}\hat{p}_x$ can be built from basic operators $\hat{x}$, $\hat{y}$, $\hat{p}_x$, and $\hat{p}_y$, each of which has an exact QTTO representation.

(a)                                      (b)

Figure 10: The representations of tensors as (a) a matrix of vectors for a QTT, and (b) a matrix of matrices for an MPO. The corresponding indices for the elements are shown in the lower panels.

## C  Exact QTT representations

Here we introduce some functions and operators that have the exact QTT representations. To represent the tensors, we write the tensors in matrix form, where each element is a vector (matrix) for a function (operator). The corresponding indices are sketched in Fig.10(a-b).

**Linear functions.**   A linear function $f(x) = ax + b$ can be represented in the QTT form efficiently. It is convenient to first consider $f(x) \to f_{b_1 b_2 \cdots b_N} = x_{\mathrm{Int}}$ for integer $x_{\mathrm{Int}}$ from 0 to $2^N - 1$. The QTT representation for $f_{b_1 b_2 \cdots b_N}$ can be easily obtained from the binary representation of integers, $x_{\mathrm{Int}} = \sum_{i=1}^{N} b_i 2^{i-1}$, where $b_i = \{0,1\}$ are the qubit degrees of freedom and $N$ is the number of qubits. The QTT tensor on site $i$ is thus

$$A(i) = \begin{pmatrix} \mathbf{I} & \mathbf{0} \\ \mathbf{t}_i & \mathbf{I} \end{pmatrix}, \quad \mathbf{I} = \begin{pmatrix} 1 \\ 1 \end{pmatrix}, \quad \mathbf{t}_i = \begin{pmatrix} 0 \\ 2^{i-1} \end{pmatrix} \tag{16}$$

– each $\mathbf{t}_i$ contributes 0 or $2^i$ to $x$ depending on $b_i$. The product of the tensors accumulate the sum of $\mathbf{t}_i$ to the corner,

$$\prod_i A(i) = \begin{pmatrix} \mathbf{I} & \mathbf{0} \\ \sum_i \mathbf{t}_i & \mathbf{I} \end{pmatrix}. \tag{17}$$

The left and the right edge tensors are defined as

$$L = \begin{pmatrix} 0 & 1 \end{pmatrix}, \quad R = \begin{pmatrix} 1 \\ 0 \end{pmatrix}. \tag{18}$$

to pick up the corner element $\sum_i \mathbf{t}_i$ from $\prod_i A(i)$.

To represent $f(x) = x$ in an arbitrary range between $x_i$ and $x_f$ with $N$ qubits (corresponding to $2^N$ descritization points), one can rescale and translate the integer $x_{\mathrm{Int}}$ to the desired range. More precisely, $f(x) = x = \alpha x_{\mathrm{Int}} + \beta$ where $\alpha = \frac{x_f - x_i}{2^N}$ and $\beta = x_i$. This can be done by defining $L$ as

$$L = \begin{pmatrix} \beta & \alpha \end{pmatrix}. \tag{19}$$

A general linear function $f(x) = ax + b$ in a range between $x_i$ and $x_f$ can be obtained by using $\alpha = a \frac{x_f - x_i}{2^N}$ and $\beta = b + x_i$.

**Polynomial functions.** A polynomial function can be obtained by using product and addition from $f(x) = x$. The product and the addition can be achieved with the standard MPS algorithms.

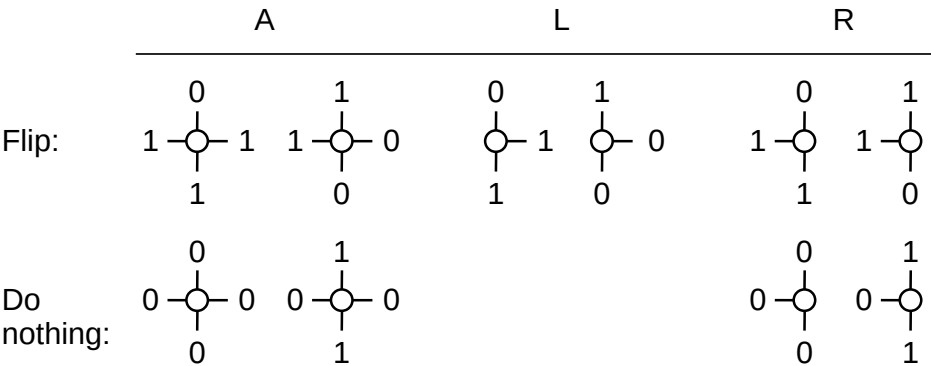

Figure 11: The non-zero elements in the tensors $A$, $L$ and $R$ for the operator $\frac{d}{dx}$. The numbers indicate the labels for the non-zero elements. These elements have values equal to one.

**Differential operators** The most important operator for us is the Laplace operator $\nabla^2$ which allows us to define the kinetic energy operator. To obtain the MPO form of $\nabla^2$, we first consider a one-dimensional differential operator $\frac{d}{dx}$. The higher dimensional operators can be obtained straightforwardly by extending the number of qubits. In the discrete space, $\frac{d}{dx} \rightarrow \frac{1}{\Delta x}\left[\hat{T}(\Delta x) - \hat{I}\right]$ where $\hat{T}(\Delta x)$ is the translational operator for a smallest distance $\Delta x$ in the discretized space, and $\hat{I}$ is the identity operator. The task is to find out $\hat{T}(\Delta x)$ defined by $\hat{T}(\Delta x)f(x) = f(x + \Delta x)$, which is nothing but an *"plus one"* operator for binary numbers. For example, $\hat{T}(\Delta x)$ maps a binary number 1100 from the lower qubits to 0010 from the upper qubits. Note that to be consistent with the programming language, we consider the smallest scale represented by the leftmost qubit. The logic of this operator can be summarized as follows. For each qubit,

1. if all the qubits from the left are 1, flip the qubit from 0 to 1 or from 1 to 0;

2. otherwise, do nothing.

This operation can be encoded in an MPO tensor with bond dimension 2, where the left index $j$ is used to indicate the above two conditions:

1. $j = 1$: All the qubits from the left are 1

2. $j = 0$: Otherwise

As a result, the tensor has only 4 non-zero elements with value 1, as shown in Fig.11, which can be concluded by the following tensor

$$A = \begin{pmatrix} \hat{I} & \hat{\sigma}^+ \\ \hat{0} & \hat{\sigma}^- \end{pmatrix} \tag{20}$$

where $\hat{I}$ is the identical matrix, and $\hat{\sigma}^+$ and $\hat{\sigma}^-$ are the raising and lowering operators. The rightmost qubit follows the same rules so the tensor is the same except no right index

is needed. The leftmost qubit (first digit) should always flip. Therefore the left and the right edge tensors are defined as

$$L = \begin{pmatrix} 0 & 1 \end{pmatrix}, \quad R = \begin{pmatrix} 1 \\ 1 \end{pmatrix}. \tag{21}$$

Note that in this definition, the derivative of the last point $11\cdots1$ is defined by the last point $11\cdots1$ and the first point $00\cdots0$. In this sense, the wavefunction has periodic boundary condition.

## D   Convergence behavior

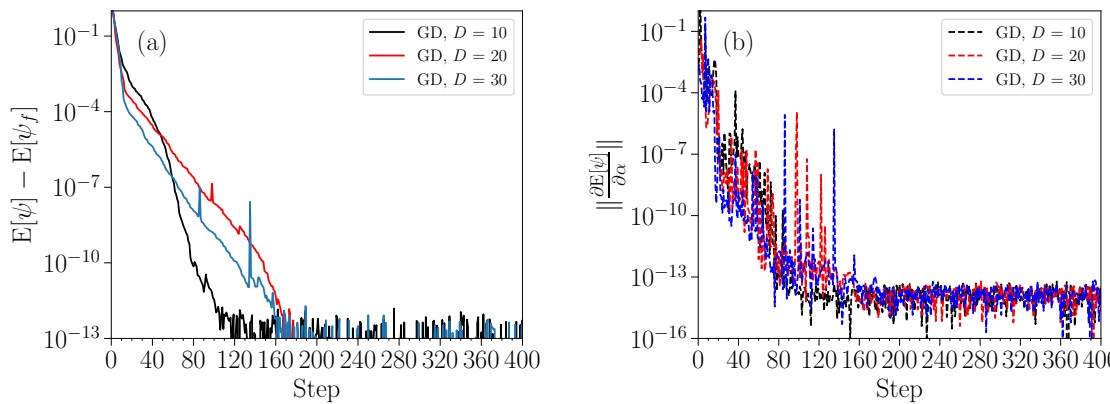

Figure 12: (a): In the GD method, the value of the variational functional $E[\psi]$ should strictly decrease with the number of steps in the early steps (Step <30). As the functional value converges to the precision of the MPS and MPO themselves ($\epsilon$: cutoff), the functional value starts to exhibit noise (Step >50). $E_{\psi_f}$ is the functional value at the last step. (b): It is the derivative of E with respect to step length $\alpha$. For the GD algorithm, determining whether $\psi$ has converged means checking if $\frac{\partial E_\psi}{\partial \alpha}$ is zero or it has reached machine precision.

Here we discuss more details about the convergence properties in the QTT methods. In the variational method, we minimize the energy $E[\psi]$ rather than the chemical potential $\mu$. Therefore, $\mu$ can be lower than the exact value during the optimizations. In Fig. 12(a), we show the convergence of the energy for the system of BEC in a harmonic trap. It can be seen that the energy decreases almost monotonically and is always higher than the exact value during the optimization steps. In some steps, the energy jumps up and then decreases again. This is because the gradient in our method is computed approximately and could sometimes allow an increase in energy. However, this does not slow down the overall convergence in the algorithm. In Fig. 12(b), we show the energy slopes along the searching directions as functions of optimization steps. Similar to the energies, the slopes decrease rapidly to almost zero with optimization steps, showing convergence of the simulations.

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
