# Peer review of "Solving the Gross-Pitaevskii Equation with Quantic Tensor Trains: Ground States and Nonlinear Dynamics"

_SciPost Physics_

## Round 1 · Referee Report · Anonymous (Referee 1) · 2025-9-29

Report

The authors propose a tensor method based on the quantic tensor train for the spatial discretization of the GPE. The computation of both ground states and dynamics is considered. Numerical experiments are presented to demonstrate the high accuracy of the proposed method, and the reduced computational cost compared to the grid based methods.

Overall, the manuscript presents a novel and interesting method. However, I have several concerns about the paper.

Main concern: The numerical examples are too simple and thus are not very convincing. From the reviewer's perspective, all the presented numerical results (including the cases of many vortices) can be obtained by using the Fourier pseudospectral method in space [2,3,4]. I would expect more examples that clearly cannot be obtained by the classical grid-based methods.

Other comments:

  1. P5. ``The time step is set to be $0.5 \delta x^2$ to satisfy the CFL condition". In fact, by using the semi-implicit discretization in the ITE (see [1]), one no longer need such CFL condition and the convergence of the ITE could be significantly accelerated.

On the other hand, many more efficient methods than the ITE are proposed for computing the ground states, e.g. [2,3,4]. Some discussion on whether the existing methods proposed for the grid-based methods can be transferred to the tensor framework would be very helpful.

  1. For the computation of dynamics. The proposed method is a spatial discretization. There are many standard temporal discretizations available in the literature, such as the splitting methods, exponential integrators, Crank-Nicolson methods, and Runge-Kutta methods. Can the proposed method be efficiently coupled with some of these temporal discretizations?

In addition, there are some challenges in the computation of dynamics for standard numerical methods, such as the simulation in the whole space instead of a truncated bounded domain, the simulation of low regularity solutions or solutions with high oscillation in space. Will the proposed methods be (potentially) advantageous in these problems?

  1. The numerical schemes in section 3, as well as the schemes used for comparison, are not clearly given in the paper, which should be revised.

References [1] W. Bao, Q. Du, Computing the ground state solution of Bose-Einstein condensates by a normalized gradient flow, SIAM J. Sci. Comput., 25 (2004), pp. 1674--1697.

[2] I. Danaila, B. Protas, Computation of Ground States of the Gross-Pitaevskii Functional via Riemannian Optimization, SIAM J. Sci. Comput., 39 (2017), pp. B1102--B1129.

[3] J. Gaidamour, Q. Tang, X. Antoine, BEC2HPC: A HPC spectral solver for nonlinear Schr\"odinger and rotating Gross-Pitaevskii equations. Stationary states computation, Comput. Phys. Commun., 265 (2021), p. 108007.

[4] H. Chen, G. Dong, W. Liu, Z. Xie, Second-order flows for computing the ground states of rotating Bose-Einstein condensates, J. Comput. Phys., 475 (2023), p. 111872.

Recommendation

Ask for major revision

  • validity: good
  • significance: high
  • originality: top
  • clarity: good
  • formatting: good
  • grammar: perfect

Author:  Chia-Min Chung  on 2025-12-10  [id 6133]

(in reply to Report 1 on 2025-09-29)
Disclosure of Generative AI use

The comment author discloses that the following generative AI tools have been used in the preparation of this comment:

AI was used to improve the English presentation in the response.

The authors propose a tensor method based on the quantic tensor train for the spatial discretization of the GPE. The computation of both ground states and dynamics is considered. Numerical experiments are presented to demonstrate the high accuracy of the proposed method, and the reduced computational cost compared to the grid based methods.

Overall, the manuscript presents a novel and interesting method. However, I have several concerns about the paper.

We thank the referee for recognizing the novelty of our paper.

Main concern: The numerical examples are too simple and thus are not very convincing. From the reviewer's perspective, all the presented numerical results (including the cases of many vortices) can be obtained by using the Fourier pseudospectral method in space [2,3,4]. I would expect more examples that clearly cannot be obtained by the classical grid-based methods.

We understand the referee’s concern. To emphasize the power of the QTT approach, we have increased the number of grids up to 2^17x2^17 and simulated up to 125 vortices. In addition, we benchmark the efficiency of our method with several standard grid-based methods that have published code, including the Fourier pseudospectral method. See Fig.7 and Fig.8 for our new benchmark results. We believe that these new results show convincing evidence for the breakthrough of the QTT approach. The new results are included in the revised manuscript.

Other comments: P5. The time step is set to be 0.5δx^2 to satisfy the CFL condition". In fact, by using the semi-implicit discretization in the ITE (see [1]), one no longer need such CFL condition and the convergence of the ITE could be significantly accelerated.

Indeed, using implicit or semi-implicit methods for time evolution can solve the constraint of a small time step. However, it is also computationally more expensive because it requires solving an equation at every time step. Whether an explicit or implicit approach is overall more efficient in this particular problem is not clear to us. Therefore, we choose to use the simple explicit scheme and focus on demonstrating the efficiency of the exponentially fine grids. To our best knowledge, the implicit method has not yet been implemented with the QTT framework. It is an interesting research direction, but it goes beyond the scope of the present study

On the other hand, many more efficient methods than the ITE are proposed for computing the ground states, e.g. [2,3,4]. Some discussion on whether the existing methods proposed for the grid-based methods can be transferred to the tensor framework would be very helpful.

As our response to Referee #2, the QTT framework can in principle be combined with existing methods. For example, we are aware of ongoing research on applying adaptive meshes to QTT. This kind of combination would allow an approach that benefits from both techniques. We added a paragraph in Sec. 5 (Conclusion) to address this perspective.

For the computation of dynamics. The proposed method is a spatial discretization. There are many standard temporal discretizations available in the literature, such as the splitting methods, exponential integrators, Crank-Nicolson methods, and Runge-Kutta methods. Can the proposed method be efficiently coupled with some of these temporal discretizations?

In principle, yes, the QTT approach can be nicely combined with other existing techniques. Specifically for dynamics, some typical discretization methods have been used with the QTT framework; for example, the Euler method [1], and the Runge-Kutta method (RK4) [2]. In our work, we use a time-evolution method called TDVP (see our Ref. [22,23]) for dynamic computation, which is commonly used in MPS applications. In principle, other methods (explicit or implicit) can also be incorporated with the QTT framework. It is interesting to explore the efficacy comparisons of different methods; however, such comparisons require careful benchmarking and go beyond the scope of our current work.

[1] Raghavendra Dheeraj Peddinti, Stefano Pisoni, Alessandro Marini, Philippe Lott, Henrique Argentieri, Egor Tiunov & Leandro Aolita, Nat. Com. Physics, 7, 135 (2024) [2] Leonhard Hölscher, Pooja Rao, Lukas Müller, Johannes Klepsch, Andre Luckow, Tobias Stollenwerk, and Frank K. Wilhelm, Phys. Rev. Research 7, 013112 (2025)

In addition, there are some challenges in the computation of dynamics for standard numerical methods, such as the simulation in the whole space instead of a truncated bounded domain, the simulation of low regularity solutions or solutions with high oscillation in space. Will the proposed methods be (potentially) advantageous in these problems?

To our knowledge, QTT is used to represent a function on a finite, bounded domain, and it is not yet clear to us how to extend it to the whole space. Regarding representability, QTT is particularly good and has been used to represent functions of high oscillation or some degree of low regularity. See Ref.[3,4] for some examples of QTT representing great details of functions with oscillations in different scales. So yes, QTT has strong potential to address the current challenges.

[3] Yuriel Núñez Fernández, Marc K. Ritter, Matthieu Jeannin, Jheng-Wei Li, Thomas Kloss, Thibaud Louvet, Satoshi Terasaki, Olivier Parcollet, Jan von Delft, Hiroshi Shinaoka, Xavier Waintal, SciPostPhys.18.3.104 (2025) [4] Marcel Niedermeier, Adrien Moulinas, Thibaud Louvet, Jose L. Lado, Xavier Waintal, arXiv:2507.04262

The numerical schemes in section 3, as well as the schemes used for comparison, are not clearly given in the paper, which should be revised.

In the original version, we implemented explicit ITE with regular finite-difference discretizations for the comparison with the QTT approach. In the revised version, we added a few more conventional methods to the comparisons (see Fig.8 in the revised version) and cited the corresponding reference (see Ref.[54]).

For the TDVP method we used for QTT time evolution, we added a few more sentences in the first paragraph of Sec. 3.1 to improve the introduction. We also cited, in addition to the original Ref.[22,23], an introductory lecture note (Ref. [83]) to help the readers who are not from the tensor network community. A detailed review of the method, however, in our opinion, would be too heavy for this paper.

References [1] W. Bao, Q. Du, Computing the ground state solution of Bose-Einstein condensates by a normalized gradient flow, SIAM J. Sci. Comput., 25 (2004), pp. 1674--1697. [2] I. Danaila, B. Protas, Computation of Ground States of the Gross-Pitaevskii Functional via Riemannian Optimization, SIAM J. Sci. Comput., 39 (2017), pp. B1102--B1129. [3] J. Gaidamour, Q. Tang, X. Antoine, BEC2HPC: A HPC spectral solver for nonlinear Schr\"odinger and rotating Gross-Pitaevskii equations. Stationary states computation, Comput. Phys. Commun., 265 (2021), p. 108007. [4] H. Chen, G. Dong, W. Liu, Z. Xie, Second-order flows for computing the ground states of rotating Bose-Einstein condensates, J. Comput. Phys., 475 (2023), p. 111872.

We thank the reviewers for bringing these references to our attention. We have extended our discussions on the numerical approaches in the introduction with more than 10 newly added references. The above references were also included.

---

## Round 1 · Referee Report · Anonymous (Referee 2) · 2025-10-2

Report

This paper provides a complete exploration of using the QTT format for solving GPE. The following things are clearly established: 1. The nonlinearity in QPE could be handled by QTT. 2. The bond dimension saturates and are moderate.

However, this paper comes from a QTT / MPS perspective. My biggest concern is that it does not seem to sufficiently acknowledge or situate itself within the long list of efforts to solve 2D and 3D Gross-Pitaevski equations, both for stationary states and dynamics, which starts from early 2000s and still continues to today. (For example, a quick search leads me to a review paper in DOI.2025 10.1137/22M1516324.)

  1. The current introduction section focus almost entirely on MPS / TT. I strongly recommend that the authors add at least one paragraph summarizing the well-established algorithmic developments for the GPE over the past two decades, such as for spatial discretizations (finite different, finite element, Fourier spectral methods, adaptive meshes etc), minimization algorithms (Riemannian optimzation, and the recent Sobolev gradient descent, just to name a few), and also the eigenvector-based perspective for stationary states.

  2. The manuscript repeatedly refers to “conventional” or “traditional” methods. I'd appreciate some clarification on this. For example, are these finite differences/finite elements implementation? How are the boundary conditions handled? Since these methods are used as benchmarks, a more detailed description is essential.

  3. I'd like to point out that a comparison to a regular finite-difference/finite-element method for arguing the efficiency of the QTT approach is not necessarily a fair comparison. There exist well-developed improvements within grid-based frameworks — e.g., multigrid solvers and adaptive meshes — which are widely used for a few decades. One could even argue that finite-element methods could deal naturally accommodate irregular geometries while it is not clear how QTT could do that.

This is not to say that the authors are obligated to do comparisons with adaptive grids / multgrid methods, but they still need to fairly address this perspective. Moreover, if they do such comparisons, this paper will be much more convincing scientifically.

  1. As a result of Point 3, I invite the authors to reexamine and reconsider some of the arguments they make, about the advantages of the QTT approach. For example, here is a paper from 2006 (DOI. 10.1137/050629392), which calculates rotating BEC with comparable number of vortices to this paper. This may motivate the authors to reevaluate how the contributions of this paper should be described. Positioning the QTT approach as one among several methods, rather than as a categorical improvement over “conventional” techniques, might be more suitable.

I want to conclude by saying that my comments are not meant to discourage the pursuit of new algorithms for classical problems. On the contrary, I believe such efforts are very valuable, provided they are put into proper context and compared carefully with existing approaches.

Recommendation

Ask for major revision

  • validity: good
  • significance: ok
  • originality: good
  • clarity: high
  • formatting: perfect
  • grammar: perfect

Author:  Chia-Min Chung  on 2025-12-10  [id 6132]

(in reply to Report 2 on 2025-10-02)
Disclosure of Generative AI use

The comment author discloses that the following generative AI tools have been used in the preparation of this comment:

AI was used to improve the English presentation in the report.

This paper provides a complete exploration of using the QTT format for solving GPE. The following things are clearly established: 1. The nonlinearity in QPE could be handled by QTT. 2. The bond dimension saturates and are moderate.

However, this paper comes from a QTT / MPS perspective. My biggest concern is that it does not seem to sufficiently acknowledge or situate itself within the long list of efforts to solve 2D and 3D Gross-Pitaevski equations, both for stationary states and dynamics, which starts from early 2000s and still continues to today. (For example, a quick search leads me to a review paper in DOI.2025 10.1137/22M1516324.)

The current introduction section focus almost entirely on MPS / TT. I strongly recommend that the authors add at least one paragraph summarizing the well-established algorithmic developments for the GPE over the past two decades, such as for spatial discretizations (finite different, finite element, Fourier spectral methods, adaptive meshes etc), minimization algorithms (Riemannian optimzation, and the recent Sobolev gradient descent, just to name a few), and also the eigenvector-based perspective for stationary states.

We appreciate the reviewer’s valuable comment. We have revised the relevant content by adding new paragraphs on page 3 in Sec. 1 (Introduction) and by citing review articles and other numerical works, with more than 10 new references. We have also extended the introduction with two additional paragraphs discussing standard algorithms and methods for both stationary and dynamical studies on page 3, including imaginary-time evolution, gradient-flow and Newton methods for stationary problems, as well as finite-difference and pseudo-spectral methods for the approaches for spatial derivatives. Common time-propagation methods are also addressed. We believe the revised version now better situates our work within the broader literature on numerical methods.

The manuscript repeatedly refers to “conventional” or “traditional” methods. I'd appreciate some clarification on this. For example, are these finite differences/finite elements implementation? How are the boundary conditions handled? Since these methods are used as benchmarks, a more detailed description is essential.

Yes, “conventional” methods refer to the methods with regular finite-difference discretizations. We now consistently use “conventional” instead of “traditional.” We added a footnote at the first occurrence of the word “conventional” to clarify its meaning.

We use periodic boundary conditions in our simulations. However, since we consider a wavefunction in a trapping potential, the wavefunction amplitude decays away from the center. We carefully choose our system size such that the wavefunction amplitude is tiny (roughly 10^-9) at the boundary. Therefore, the boundary condition does not have a significant effect on our problem. We included this information in our revised version.

I'd like to point out that a comparison to a regular finite-difference/finite-element method for arguing the efficiency of the QTT approach is not necessarily a fair comparison. There exist well-developed improvements within grid-based frameworks — e.g., multigrid solvers and adaptive meshes — which are widely used for a few decades. One could even argue that finite-element methods could deal naturally accommodate irregular geometries while it is not clear how QTT could do that.

We understand the referee’s concern. To emphasize the power of our QTT approach, we have increased the number of grids up to 2^17x2^17 and simulated up to 125 vortices, which is very challenging for the regular finite-difference methods. In addition, we benchmark the efficiency of our method with several standard grid-based methods that have published code, as shown in Fig.7 and Fig.8 in the revised version. The new simulations are very challenging when using regular grid-based methods, as indicated by the slow convergence in the figure, demonstrating the fundamental breakthrough of the QTT approach in the fine-grid limit. These new benchmark results are included in the revised version of the manuscript.

We also note that the QTT framework targets a specific type of approximation and, in principle, can be combined with existing methods. We are aware of ongoing research on applying adaptive meshes to QTT, which could allow a combined approach that benefits from both techniques. Therefore, QTT should not be viewed simply as competing with other methods, but rather as an alternative approach that can be effectively integrated with them. Exploring such combinations is an interesting direction for future work, but it is beyond the scope of the present study.

This is not to say that the authors are obligated to do comparisons with adaptive grids / multgrid methods, but they still need to fairly address this perspective. Moreover, if they do such comparisons, this paper will be much more convincing scientifically.

We thank the referee for the valuable suggestions.

As a result of Point 3, I invite the authors to reexamine and reconsider some of the arguments they make, about the advantages of the QTT approach. For example, here is a paper from 2006 (DOI. 10.1137/050629392), which calculates rotating BEC with comparable number of vortices to this paper. This may motivate the authors to reevaluate how the contributions of this paper should be described. Positioning the QTT approach as one among several methods, rather than as a categorical improvement over “conventional” techniques, might be more suitable.

Following our response to the referee’s last comment and the inclusion of new results, we believe we have provided convincing evidence that the QTT approach represents a fundamental breakthrough. This new framework can be further integrated with existing methods. To clarify this point for readers, we revised the manuscript to explain its relation to other approaches better.

I want to conclude by saying that my comments are not meant to discourage the pursuit of new algorithms for classical problems. On the contrary, I believe such efforts are very valuable, provided they are put into proper context and compared carefully with existing approaches.

We thank the referee for the comments and for carefully examining the value of our work.

---

## Round 1 · Referee Report · Anonymous (Referee 3) · 2025-10-27

Disclosure of Generative AI use

The referee discloses that the following generative AI tools have been used in the preparation of this report:

Used for Grammar correction, but I gave the original content and information.

Strengths

  • The method tackles a nonlinear quantum PDE with broad applications in physics.
  • The QTT framework allows exponential compression, enabling simulations with high resolution at reduced computational cost.
  • The variational method shows superior performance over TDVP in convergence, which is essential for nonlinear solvers.
  • Benchmark results include challenging systems (e.g., 61-vortex states), which suggests scalability.

Weaknesses

  • The comparison is only made with the most naive baseline solver. While this is understandable at this early stage of development, the authors should clarify this limitation explicitly in the text. As Referee 2 points out, this method is not yet compared to the current state-of-the-art. However, it is reasonable not to expect a newly proposed QTT approach to outperform highly optimized traditional solvers with a long development history.

Report

The manuscript "Solving the Gross–Pitaevskii Equation with Quantic Tensor Trains: Ground States and Nonlinear Dynamics" proposes a framework to solve the nonlinear Schrödinger-type Gross–Pitaevskii equation using Quantic Tensor Trains (QTT). The approach combines the time-dependent variational principle (TDVP) and variational optimization and demonstrates strong performance for complex configurations, including systems with 61 vortices.

The results are timely and technically solid. The convergence advantages of the variational method over TDVP are particularly interesting for nonlinear solvers. The method could stimulate further work in efficient representations of high-dimensional nonlinear PDEs in quantum physics.

I recommend publication after minor revisions.

Requested changes

Below are detailed suggestions:

  • In Fig. 2, the vertical axis is labeled as an error, but it is unclear whether this represents absolute or relative error. Please clarify.
  • In Fig. 3, it would help readability to add a note like "See also Fig. 7 for the 7, 19, 37 vortex density plots".
  • In Fig. 4, while the QTT-based scaling is visually compared to grid-based methods, a brief note on the number of discretization points (qubits) needed for sufficient accuracy would make the comparison more balanced.
  • In Fig. 4’s legend, the bond dimension D should be italicized for consistency.
  • Appendix A is not cited in the main text. It discusses the role of normalization in the variational method, but it is unclear whether strict normalization is essential or whether the observed difficulties also appear without normalization.

Technical definitions (should be clarified in the revised manuscript): - The QTT compression error is defined as 1 - |<ψ|ψ_QTT>|, but a more geometric error would be |1 - <ψ|ψ_QTT>|, which satisfies ‖ψ - ψ_QTT‖ ~ O(δ^2). It would help to comment on this, as some tensor network literature (e.g., ITensor) uses the latter form for truncation criteria. - In Fig. 8, the error measure ε_ψ² is not defined in the caption or main text. It should be stated explicitly whether it refers to the Frobenius norm ‖ψ - ψ_QTT‖_F or its square.

Recommendation

Publish (easily meets expectations and criteria for this Journal; among top 50%)

  • validity: good
  • significance: good
  • originality: high
  • clarity: good
  • formatting: good
  • grammar: good

Author:  Chia-Min Chung  on 2025-12-10  [id 6131]

(in reply to Report 3 on 2025-10-27)
Disclosure of Generative AI use

The comment author discloses that the following generative AI tools have been used in the preparation of this comment:

AI was used to improve the English presentation in this report.

The manuscript "Solving the Gross–Pitaevskii Equation with Quantic Tensor Trains: Ground States and Nonlinear Dynamics" proposes a framework to solve the nonlinear Schrödinger-type Gross–Pitaevskii equation using Quantic Tensor Trains (QTT). The approach combines the time-dependent variational principle (TDVP) and variational optimization and demonstrates strong performance for complex configurations, including systems with 61 vortices.

The results are timely and technically solid. The convergence advantages of the variational method over TDVP are particularly interesting for nonlinear solvers. The method could stimulate further work in efficient representations of high-dimensional nonlinear PDEs in quantum physics.

We thank the referee for the positive evaluation and for recognizing the significance and potential impact of our work.

I recommend publication after minor revisions. Requested changes Below are detailed suggestions: In Fig. 2, the vertical axis is labeled as an error, but it is unclear whether this represents absolute or relative error. Please clarify. We thank the referee for pointing this out.

They are absolute errors. We will clarify this in the caption of Fig.2 in the revised version.

In Fig. 3, it would help readability to add a note like "See also Fig. 7 for the 7, 19, 37 vortex density plots".

We thank the referee for the suggestion. We referred to the corresponding figure in the caption of Fig.3 in the revised manuscript.

In Fig. 4, while the QTT-based scaling is visually compared to grid-based methods, a brief note on the number of discretization points (qubits) needed for sufficient accuracy would make the comparison more balanced.

We included a new Fig.13 in Appendix D in the revised version to show the convergence of the energy and chemical potential as functions of grid size.

In Fig. 4’s legend, the bond dimension D should be italicized for consistency.

We thank the referee for the careful reading. We have fixed it in the revised version.

Appendix A is not cited in the main text. It discusses the role of normalization in the variational method, but it is unclear whether strict normalization is essential or whether the observed difficulties also appear without normalization.

We thank the referee for pointing this out. We have cited Appendix A at the end of the first paragraph in Sec.2

Technical definitions (should be clarified in the revised manuscript): - The QTT compression error is defined as 1 - |<ψ|ψ_QTT>|, but a more geometric error would be |1 - <ψ|ψ_QTT>|, which satisfies ‖ψ - ψ_QTT‖ ~ O(δ^2). It would help to comment on this, as some tensor network literature (e.g., ITensor) uses the latter form for truncation criteria. - In Fig. 8, the error measure ε_ψ² is not defined in the caption or main text. It should be stated explicitly whether it refers to the Frobenius norm ‖ψ - ψ_QTT‖_F or its square.

We understand the referee’s concern. However, the error we are using has a nice property that it has a value between 0 (when two states are the same up to a phase) and 1 (when two states are orthogonal). Although the geometric error is also commonly used, its value is between 0 and 2, and it has a value of 2 when psi_QTT = -psi, which is considered a perfect approximation in our case. We therefore would prefer to keep the property of our error definition. To make our error definition more conventional, we define our error as the Fidelity error, 1 - |<psi|psi_QTT>|^2, where |<psi|psi_QTT>|^2 is understood as the fidelity, in the revised version.

---

## Round 3 · Author Response

We thank you and the three reviewers for the careful reading of our manuscript “Solving the Gross–Pitaevskii Equation with Quantic Tensor Trains: Ground States and Nonlinear Dynamics” and for the constructive comments and suggestions. We are pleased that all reviewers found the topic timely and the approach technically solid. We have carefully revised the manuscript in response to all comments, and we believe the new version has been substantially improved in clarity, scope, and scientific positioning.
Below we provide a detailed, point-by-point response to the reviewers’ reports, summarizing all revisions implemented in the manuscript.

---

## Round 3 · List of Changes

Major changes: 1. We add a new figure, Fig. 7, for new simulations with a much finer 2^17×2^17 discretization, and up to 125 vortices. Fig.7 shows the vortex density profiles for the systems with different numbers of vortices. 2. We add a new figure, Fig. 8, showing a benchmark of the efficiency of our method compared with several regular finite-difference methods. The system has 7 vortices (see Fig. 7(a)) on a 2^11×2^11 grid. 3. We add a new paragraph in Sec. 4.2 discussing our new benchmark results above. In Sec. 1, Introduction, we add a new paragraph, expand the other corresponding paragraphs, and cite corresponding references to summarize the existing methods for solving the GPE.
Other changes: 4. We update Table 1 for the parameters used in the new simulations. 5. We add Eq. 4 to clearly state the normalization condition we use and refer readers to Appendix A for more details. 6. In Sec. 3, we add a new paragraph indicating the boundary condition and explain why the boundary effect is negligible in our systems. 7. In Sec. 3.1, we add a few sentences and cite an introductory lecture note (Ref. 58) for a more thorough introduction to the TDVP method. 8. In the caption of Fig.2, we clearly state that the error is the absolute error. 9. In Fig.3, we change the definition of the error to the fidelity error 1-|<psipsi_QTT>|^2, and state it in the main text. 10. We add Fig. 13 in Appendix D to discuss convergence with the number of qubits (grid resolution).

---

## Editorial Decision

unknown